# SUBMODULAR SUM-PRODUCT NETWORKS FOR SCENE UNDERSTANDING

**Abram L. Friesen & Pedro Domingos**
Department of Computer Science and Engineering
University of Washington
Seattle, WA 98195, USA
`{afriesen,pedrod}@cs.washington.edu`

## ABSTRACT

Sum-product networks (SPNs) are an expressive class of deep probabilistic models in which inference takes time linear in their size, enabling them to be learned effectively. However, for certain challenging problems, such as scene understanding, the corresponding SPN has exponential size and is thus intractable. In this work, we introduce submodular sum-product networks (SSPNs), an extension of SPNs in which sum-node weights are defined by a submodular energy function. SSPNs combine the expressivity and depth of SPNs with the ability to efficiently compute the MAP state of a combinatorial number of labelings afforded by submodular energies. SSPNs for scene understanding can be understood as representing all possible parses of an image over *arbitrary region shapes* with respect to an image grammar. Despite this complexity, we develop an efficient and convergent algorithm based on graph cuts for computing the (approximate) MAP state of an SSPN, greatly increasing the expressivity of the SPN model class. Empirically, we show exponential improvements in parsing time compared to traditional inference algorithms such as $\alpha$-expansion and belief propagation, while returning comparable minima.

## 1   INTRODUCTION

Sum-product networks (SPNs) (Poon & Domingos, 2011; Gens & Domingos, 2012) are a class of deep probabilistic models that consist of many layers of hidden variables and can have unbounded treewidth. Despite this depth and corresponding expressivity, exact inference in SPNs is guaranteed to take time linear in their size, allowing their structure and parameters to be learned effectively from data. However, there are still many models for which the corresponding SPN has size exponential in the number of variables and is thus intractable. For example, in scene understanding (or semantic segmentation), the goal is to label each pixel of an image with its semantic class, which requires simultaneously detecting, segmenting, and recognizing each object in the scene. Even the simplest SPN for scene understanding is intractable, as it must represent the exponentially large set of segmentations of the image into its constituent objects.

Scene understanding is commonly formulated as a flat Markov (or conditional) random field (MRF) over the pixels or superpixels of an image (e.g., Shotton et al. (2006); Gould et al. (2009)). Inference in MRFs is intractable in general; however, there exist restrictions of the MRF that enable tractable inference. For pairwise binary MRFs, if the energy of each pairwise term is submodular (alternatively, attractive or regular) (Kolmogorov & Zabih, 2004), meaning that each pair of neighboring pixels prefers to have the same label, then the exact MAP labeling of the MRF can be recovered in low-order polynomial time through the use of a graph cut algorithm[1] (Greig et al., 1989; Boykov & Kolmogorov, 2004). This result from the binary case has been used to develop a number of powerful approximate algorithms for the multi-label case (e.g., Komodakis et al. (2007); Lempitsky et al. (2010)), the most well-known of which is $\alpha$-expansion (Boykov et al., 2001), which efficiently returns an approximate labeling that is within a constant factor of the true optimum by solving a series of binary graph cut problems. Unfortunately, pairwise MRFs are insufficiently expressive for com-

---

[1] Formally, a min-cut/max-flow algorithm(Ahuja et al., 1993) on a graph constructed from the MRF.

plex tasks such as scene understanding, as they are unable to model high-level relationships, such as constituency (part-subpart) or subcategorization (superclass-subclass), between arbitrary regions of the image, unless these can be encoded in the labels of the MRF and enforced between pairs of (super)pixels. However, this encoding requires a combinatorial number of labels, which is intractable. Instead, higher-level structure is needed to efficiently represent these relationships.

In this paper, we present *submodular sum-product networks* (SSPNs), a novel model that combines the expressive power of sum-product networks with the tractable segmentation properties of submodular energies. An SSPN is a sum-product network in which the weight of each child of a sum node corresponds to the energy of a particular labeling of a submodular energy function. Equivalently, an SSPN over an image corresponds to an instantiation of all possible parse trees of that image with respect to a given image grammar, where the probability distribution over the segmentations of a production on a particular region is defined by a submodular random field over the pixels in that region. Importantly, SSPNs permit objects and regions to take *arbitrary shapes*, instead of restricting the set of possible shapes as has previously been necessary for tractable inference. By exploiting submodularity, we develop a highly-efficient approximate inference algorithm, INFERSSPN, for computing the MAP state of the SSPN (equivalently, the optimal parse of the image). INFERSSPN is an iterative move-making-style algorithm that provably converges to a local minimum of the energy, reduces to $\alpha$-expansion in the case of a trivial grammar, and has complexity $O(|G|c(n))$ for each iteration, where $c(n)$ is the complexity of a single graph cut and $|G|$ is the size of the grammar. As with other move-making algorithms, INFERSSPN converges to a local minimum with respect to an exponentially-large set of neighbors, overcoming many of the main issues of local minima (Boykov et al., 2001). Empirically, we compare INFERSSPN to belief propagation (BP) on a multilevel MRF and to $\alpha$-expansion on an equivalent flat MRF. We show that INFERSSPN parses images in exponentially less time than both of these while returning energies comparable to $\alpha$-expansion, which is guaranteed to return energies within a constant factor of the true optimum.

The literature on using higher-level information for scene understanding is vast. We briefly discuss the most relevant work on hierarchical random fields over multiple labels, image grammars for segmentation, and neural parsing methods. Hierarchical random field models (e.g., Russell et al. (2010); Lempitsky et al. (2011)) define MRFs with multiple layers of hidden variables and then perform inference, often using graph cuts to efficiently extract the MAP solution. However, these models are typically restricted to just a few layers and to pre-computed segmentations of the image, and thus do not allow arbitrary region shapes. In addition, they require a combinatorial number of labels to encode complex grammar structures. Previous grammar-based methods for scene understanding, such as Zhu & Mumford (2006) and Zhao & Zhu (2011), have used MRFs with AND-OR graphs (Dechter & Mateescu, 2007), but needed to restrict their grammars to a very limited set of productions and region shapes in order to perform inference in reasonable time, and are thus much less expressive than SSPNs. Finally, neural parsing methods such as those in Socher et al. (2011) and Sharma et al. (2014) use recursive neural network architectures over superpixel-based features to segment an image; thus, these methods also do not allow arbitrary region shapes. Further, Socher et al. (2011) greedily combine regions to form parse trees, while (Sharma et al., 2014) use randomly generated parse trees, whereas inference in SSPNs finds the (approximately) optimal parse tree.

## 2 SUBMODULAR SUM-PRODUCT NETWORKS

In the following, we define submodular sum-product networks (SSPNs) in terms of an image grammar because this simplifies the exposition with respect to the structure of the sum-product network (SPN) and because scene understanding is the domain we use to evaluate SSPNs. However, it is not necessary to define SSPNs in this way, and our results extend to any SPN with sum-node weights defined by a random field with submodular potentials. Due to lack of space we refer readers to Gens & Domingos (2012), Poon & Domingos (2011) and Gens & Domingos (2013) for SPN details.

With respect to scene understanding, an SSPN defines a generative model of an image and a hierarchy of regions within that image where each region is labeled with a production (and implicitly by the head symbol of that production), can have arbitrary shape, and is a subset of the region of each of its ancestors. An example of an SSPN for parsing a farm scene is shown in Figure 1. Given a starting symbol and the region containing the entire image, the generative process is to first choose a production of that symbol into its constituent symbols and then choose a segmentation of the region into a set of mutually exclusive and exhaustive subregions, with one subregion per constituent sym-

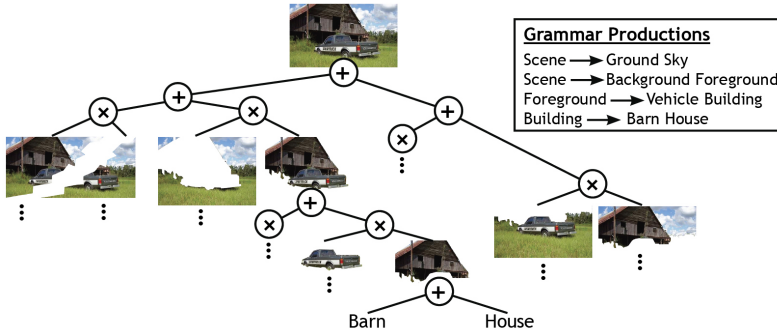

Figure 1: A partial (submodular) sum-product network for parsing an image with respect to the grammar shown. There is a sum node for each nonterminal symbol with a child sum node for each production of that symbol. Each sum node for a production has a child product node for each possible segmentation of its region.

bol. The process then recurses, choosing a production and a segmentation for each subregion given its symbol. The recursion terminates when one of the constituents is a terminal symbol, at which point the pixels corresponding to that region of the image are generated. This produces a parse tree in which each internal node is a pair containing a region and a production of the region, and the leaves are regions of pixels. For each node in a parse tree, the regions of its children are mutually exclusive and exhaustive with respect to the parent node's region. As in a probabilistic context-free grammar (PCFG) (Jurafsky & Martin, 2000), productions are chosen from a categorical distribution over the productions of the current symbol. Segmentations of a given region, however, are sampled from a (submodular) Markov random field (MRF) over the pixels in the region.

Formally, let $G = (N, \Sigma, R, S, \mathbf{w})$ be a non-recursive stochastic grammar, where $N$ is a finite set of nonterminal symbols; $\Sigma$ is a finite set of terminal symbols; $R$ is a finite set of productions $R = \{v : X \to Y_1 Y_2 \ldots Y_k\}$ with head symbol $X \in N$ and constituent symbols $Y_i \in N \cup \Sigma$ for $i = 1 \ldots k$ and $k > 0$; $S \in N$ is a distinguished start symbol, meaning that it does not appear on the right-hand side of any production; and $\mathbf{w}$ are the weights that parameterize the probability distribution defined by $G$. For a production $v \in t$ in a parse tree $t \in \mathcal{T}_G$, we denote its region as $\mathcal{P}_v$ and its parent and children as pa($v$) and ch($v$), respectively, where $\mathcal{T}_G$ is the set of possible parse trees under the grammar $G$. The labeling corresponding to the segmentation of the pixels in $\mathcal{P}_v$ for production $v : X \to Y_1 \ldots Y_k$ is $\mathbf{y}^v \in \mathcal{Y}_v^{|\mathcal{P}_v|}$, where $\mathcal{Y}_v = \{Y_1, \ldots, Y_k\}$. The region of any production $v \in t$ is the set of pixels in $\mathcal{P}_{\text{pa}(v)}$ whose assigned label is the head of $v$, i.e., $\mathcal{P}_v = \{p \in \mathcal{P}_{\text{pa}(v)} : y_p^{\text{pa}(v)} = \text{head}(v)\}$, except for the production of the start symbol, which has the entire image as its region. The probability of an image $\mathbf{x}$ is $p_{\mathbf{w}}(\mathbf{x}) = \sum_{t \in \mathcal{T}_G} p_{\mathbf{w}}(t, \mathbf{x})$, where the joint probability of parse tree $t$ and the image is the product over all productions in $t$ of the probability of choosing that production $v$ and then segmenting its region $\mathcal{P}_v$ according to $\mathbf{y}^v$:

$$p_{\mathbf{w}}(t, \mathbf{x}) = \frac{1}{Z} \exp(-E_{\mathbf{w}}(t, \mathbf{x})) = \frac{1}{Z} \exp(-\sum_{v \in t} E_{\mathbf{w}}^v(v, \mathbf{y}^v, \text{head}(v), \mathcal{P}_v, \mathbf{x})).$$

Here, $Z = \sum_{t \in \mathcal{T}_G} \exp(-E_{\mathbf{w}}(t, \mathbf{x}))$ is the partition function, $\mathbf{w}$ are the model parameters, and $E$ is the energy function. In the following, we will simplify notation by omitting head($v$), $\mathcal{P}_v$, $\mathbf{x}$, $\mathbf{w}$, and superscript $v$ from the energy function when they are clear from context. The energy of a production and its segmentation on the region $\mathcal{P}_v$ are given by a pairwise Markov random field (MRF) as $E(v, \mathbf{y}^v) = \sum_{p \in \mathcal{P}_v} \theta_p^v(y_p^v; \mathbf{w}) + \sum_{(p,q) \in \mathcal{E}_v} \theta_{pq}^v(y_p^v, y_q^v; \mathbf{w})$, where $\theta_p^v$ and $\theta_{pq}^v$ are the unary and pairwise costs of the segmentation MRF, $\{y_p^v : p \in \mathcal{P}_v\}$ is the labeling defining the segmentation of the pixels in the current region, and $\mathcal{E}_v$ are the edges in $\mathcal{P}_v$. Without loss of generality we assume that $\mathcal{E}_v$ contains only one of $(p, q)$ or $(q, p)$, since the two terms can always be combined. Here, $\theta_p^v$ is the per-pixel data cost and $\theta_{pq}^v$ is the boundary term, which penalizes adjacent pixels within the same region that have different labels. We describe these terms in more detail below. In general, even computing the segmentation for a single production is intractable. In order to permit efficient inference, we require that $\theta_{pq}^v$ satisfies the submodularity condition $\theta_{pq}^v(Y_1, Y_1) + \theta_{pq}^v(Y_2, Y_2) \leq \theta_{pq}^v(Y_1, Y_2) + \theta_{pq}^v(Y_2, Y_1)$ for all productions $v : X \to Y_1 Y_2$ once the grammar has been converted to a grammar in which each production has only two constituents, which is always possible and in the worst case increases the grammar size quadratically (Jurafsky & Martin, 2000; Chomsky,

1959). We also require for every production $v \in R$ and for every production $c$ that is a descendant of $v$ in the grammar that $\theta_{pq}^v(y_p^v, y_q^v) \geq \theta_{pq}^c(y_p^c, y_q^c)$ for all possible labelings $(y_p^v, y_q^v, y_p^c, y_q^c)$, where $y_p^v, y_q^v \in \mathcal{Y}_v$ and $y_p^c, y_q^c \in \mathcal{Y}_c$. This condition ensures that segmentations for higher-level productions are submodular, no matter what occurs below them. It also encodes the reasonable assumption that higher-level abstractions are separated by stronger, shorter boundaries (relative to their size), while lower-level objects are more likely to be composed of smaller, more intricately-shaped regions.

The above model defines a sum-product network containing a sum node for each possible region of each nonterminal, a product node for each segmentation of each production of each possible region of each nonterminal, and a leaf function on the pixels of the image for each possible region of the image for each terminal symbol. The children of the sum node $s$ for nonterminal $X_s$ with region $\mathcal{P}_s$ are all product nodes $r$ with a production $v_r : X_s \rightarrow Y_1 \ldots Y_k$ and region $\mathcal{P}_{v_r} = \mathcal{P}_s$. Each product node corresponds to a labeling $\mathbf{y}^{v_r}$ of $\mathcal{P}_{v_r}$ and the edge to its parent sum node has weight $\exp(-E(v, \mathbf{y}^{v_r}, \mathcal{P}_{v_r}))$. The children of product node $r$ are the sum or leaf nodes with matching regions that correspond to the constituent nonterminals or terminals of $v_r$, respectively. Since the weights of the edges from a sum node to its children correspond to submodular energy functions, we call this a submodular sum-product network (SSPN).

A key benefit of SSPNs in comparison to previous grammar-based approaches is that regions can have arbitrary shapes and are not restricted to a small class of shapes such as rectangles (Poon & Domingos, 2011; Zhao & Zhu, 2011). This flexibility is important when parsing images, as real-world objects and abstractions can take any shape, but it comes with a combinatorial explosion of possible parses. However, by exploiting submodularity, we are able to develop an efficient inference algorithm for SSPNs, allowing us to efficiently parse images into a hierarchy of arbitrarily-shaped regions and objects, yielding a very expressive model class. This efficiency is despite the size of the underlying SSPN, which is in general far too large to explicitly instantiate.

## 2.1 MRF SEGMENTATION DETAILS

As discussed above, the energy of each segmentation of a region for a given production is defined by a submodular MRF $E(v, \mathbf{y}^v) = \sum_{p \in \mathcal{P}_v} \theta_p^v(y_p^v; \mathbf{w}) + \sum_{(p,q) \in \mathcal{E}_v} \theta_{pq}^v(y_p^v, y_q^v; \mathbf{w})$. The unary terms in $E(v, \mathbf{y}^v)$ differ depending on whether the label $y_p^v$ corresponds to a terminal or nonterminal symbol. For a terminal $T \in \Sigma$, the unary terms are a linear function of the image features $\theta_p^v(y_p^v = T; \mathbf{w}) = w_v^{\text{PC}} + \mathbf{w}_T^\top \phi_p^U$, where $w_v^{PC}$ is an element of $\mathbf{w}$ that specifies the cost of $v$ relative to other productions and $\phi_p^U$ is a feature vector representing the local appearance of pixel $p$. In our experiments, $\phi_p^U$ is the output of a deep neural network. For labels corresponding to a nonterminal $X \in N$, the unary terms are $\theta_p^v(y_p^v = X; \mathbf{w}) = w_v^{\text{PC}} + \theta_p^c(y_p^c)$, where $c$ is the child production of $v$ in the current parse tree that contains $p$, such that $p \in \mathcal{P}_c$. This dependence makes inference challenging, because the choice of children in the parse tree itself depends on the region that is being parsed as $X$, which depends on the segmentation this unary is being used to compute.

The pairwise terms in $E(v, \mathbf{y}^v)$ are a recursive version of the standard contrast-dependent pairwise boundary potential (e.g., Shotton et al. (2006)) defined for each production $v$ and each pair of adjacent pixels $p, q$ as $\theta_{pq}^v(y_p^v, y_q^v; \mathbf{w}) = w_v^{\text{BF}} \exp(-\beta^{-1} ||\phi_p^B - \phi_q^B||^2) \cdot [y_p^v \neq y_q^v] + \theta_{pq}^c(y_p^c, y_q^c; \mathbf{w})$, where $\beta$ is half the average image contrast between all adjacent pixels in an image, $w_v^{\text{BF}}$ is the boundary factor that controls the relative cost of this term for each production, $\phi_p^B$ is the pairwise per-pixel feature vector, $c$ is the same as in the unary term above, and $[\cdot]$ is the indicator function, which has value 1 when its argument is true and is 0 otherwise. For each pair of pixels $(p, q)$, only one such term will ever be non-zero, because once two pixels are labeled differently at a node in the parse tree, they are placed in separate subtrees and thus never co-occur in any region below the current node. In our experiments, $\phi_p^B$ are the intensity values for each pixel.

## 3 INFERENCE

Scene understanding (or semantic segmentation) requires labeling each pixel of an image with its semantic class. By constructing a grammar containing a set of nonterminals in one-to-one correspondence with the semantic labels and only allowing these symbols to produce terminals, we can recover the semantic segmentation of an image from a parse tree for this grammar. In the simplest case, a grammar need contain only one additional production from the start symbol to all other nonterminals. More generally, however, the grammar encodes rich structure about the relationships

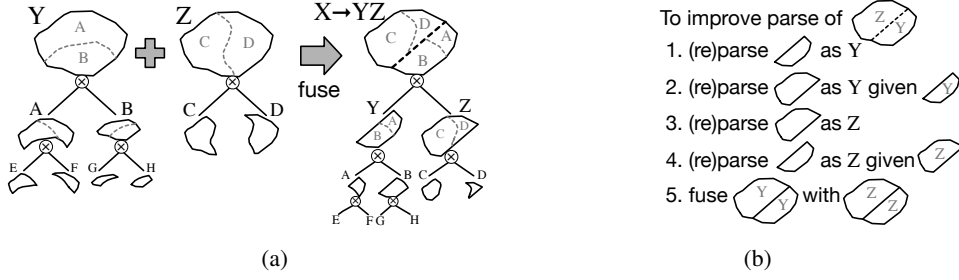

Figure 2: The two main components of INFERSSPN: (a) Parsing a region $\mathcal{P}$ as $X \to YZ$ by fusing two parses of $\mathcal{P}$ as $Y \to AB$ and as $Z \to CD$, and (b) Improving the parse of $\mathcal{P}$ as $X \to YZ$ by (re)parsing each of its subregions, taking the union of the new $Y$ and $Z$ parses of $\mathcal{P}$, and then fusing these new parses.

between image regions at various levels of abstraction, including concepts such as composition and subcategorization. Identifying the relevant structure and relationships for a particular image entails finding the best parse of an image $\mathbf{x}$ given a grammar $G$ (or, equivalently, performing MAP inference in the corresponding SSPN), i.e., $t^* = \arg\max_{t \in \mathcal{T}_G} p(t|\mathbf{x}) = \arg\min_{t \in \mathcal{T}_G} \sum_{v \in t} E(v, \mathbf{y}^v, \mathbf{x})$.

In PCFGs over sentences (Jurafsky & Martin, 2000), the optimal parse can be recovered exactly in time $O(n^3|G|)$ with the CYK algorithm (Hopcroft & Ullman, 1979), where $n$ is the length of the sentence and $|G|$ is the number of productions in the grammar, by iterating over all possible split points of the sentence and using dynamic programming to avoid recomputing sub-parses. Unfortunately, for images and other 2-D data types, there are $2^n$ possible segmentations of the data for each binary production, rendering this approach infeasible in general. With an SSPN, however, it is possible to efficiently compute the approximate optimal parse of an image. In our algorithm, INFERSSPN, this is done by iteratively constructing parses of different regions in a bottom-up fashion.

## 3.1 PARSE TREE CONSTRUCTION

Given a production $v : X \to Y_1 Y_2$ and two parse trees $t_1, t_2$ over the same region $\mathcal{P}$ and with head symbols $Y_1, Y_2$, respectively, then for any labeling $\mathbf{y}^v \in \{Y_1, Y_2\}^{|\mathcal{P}|}$ of $\mathcal{P}$ we can construct a third parse tree $t_X$ over region $\mathcal{P}$ with root production $v$, labeling $\mathbf{y}^v$, and subtrees $t_1', t_2'$ over regions $\mathcal{P}_1, \mathcal{P}_2$, respectively, such that $\mathcal{P}_i = \{p \in \mathcal{P} : y_p^v = Y_i\}$ and $t_i' = t_i \cap \mathcal{P}_i$ for each $i$, where the intersection of a parse tree and a region $t \cap \mathcal{P}$ is the new parse tree resulting from intersecting $\mathcal{P}$ with the region at each node in $t$. Of course, the quality of the resulting parse tree, $t_X$, depends on the particular labeling (segmentation) $\mathbf{y}^v$ used. Recall that a parse tree $t$ on region $\mathcal{P}$ has energy $E(t, \mathcal{P}) = \sum_{v \in t} E(v, \mathbf{y}^v, \mathcal{P}_v)$, which can be written as $E(t, \mathcal{P}) = \sum_{p \in \mathcal{P}} \theta_p^t + \sum_{(p,q) \in \mathcal{E}} \theta_{pq}^t$, where $\theta_p^t = \sum_{v \in t} \theta_p^v(y_p^v) \cdot [p \in \mathcal{P}_v]$ and $\theta_{pq}^t = \sum_{v \in t} \theta_{pq}^v(y_p^v, y_q^v) \cdot [(p,q) \in \mathcal{E}_v]$. This allows us to define the *fusion* operation, which is a key subroutine in INFERSSPN. Note that $\delta_{ij}$ is the Kronecker delta.

**Definition 1.** *For a production $v : X \to Y_1, Y_2$ and two parse trees $t_1, t_2$ over region $\mathcal{P}$ with head symbols $Y_1, Y_2$ then $t_X$ is the* fusion *of $t_1$ and $t_2$ constructed from the minimum energy labeling $\mathbf{y}^v = \arg\min_{\mathbf{y} \in \mathcal{Y}_v^{|\mathcal{P}|}} E(v, t_1, t_2, \mathbf{y})$, where*

$$
E(v, t_1, t_2, \mathbf{y}) = \sum_{p \in \mathcal{P}} \theta_p^{t_1} \cdot \delta_{y_p Y_1} + \theta_p^{t_2} \cdot \delta_{y_p Y_2} + \sum_{(p,q) \in \mathcal{E}} \theta_{pq}^{t_1} \cdot \delta_{y_p Y_1} \cdot \delta_{y_q Y_1}
$$
$$
+ \theta_{pq}^{t_2} \cdot \delta_{y_p Y_2} \cdot \delta_{y_q Y_2} + \theta_{pq}^v(Y_1, Y_2) \cdot \delta_{y_p Y_1} \cdot \delta_{y_q Y_2}.
$$

Figure 2a shows an example of fusing two parse trees to create a new parse tree. Although fusion requires finding the optimal labeling from an exponentially large set, the energy is submodular and can be efficiently optimized with a single graph cut. All proofs are presented in the appendix.

**Proposition 1.** *The energy $E(v, t_1, t_2, \mathbf{y}^v)$ of the fusion of parse trees $t_1, t_2$ over region $\mathcal{P}$ with head symbols $Y_1, Y_2$ for a production $v : X \to Y_1 Y_2$ is submodular.*

Once a parse tree has been constructed, INFERSSPN then improves that parse tree on subsequent iterations. The following result shows how INFERSSPN can improve a parse tree while ensuring that the energy of that parse tree never gets worse.

**Lemma 1.** *Given a labeling $\mathbf{y}^v$ which fuses parse trees $t_1, t_2$ into $t$ with root production $v$, energy $E(t, \mathcal{P}) = E(v, t_1, t_2, \mathbf{y}^v)$, and subtree regions $\mathcal{P}_1 \cap \mathcal{P}_2 = \emptyset$ defined by $\mathbf{y}^v$, then any improvement*

$\Delta$ *in* $E(t_1, \mathcal{P}_1)$ *also improves* $E(t, \mathcal{P})$ *by at least* $\Delta$, *regardless of any change in* $E(t_1, \mathcal{P} \backslash \mathcal{P}_1)$.

Finally, it will be useful to define the union $t = t_1 \cup t_2$ of two parse trees $t_1, t_2$ that have the same production at their root but are over disjoint regions $\mathcal{P}_1 \cap \mathcal{P}_2 = \emptyset$, as the parse tree $t$ with region $\mathcal{P} = \mathcal{P}_1 \cup \mathcal{P}_2$ and in which all nodes that co-occur in both $t_1$ and $t_2$ (i.e., have the same path to them from the root and have the same production) are merged to form a single node in $t$. In general, $t$ may be an inconsistent parse tree, as the same symbol may be parsed as two separate productions, in which case we define the energy of the boundary terms between the pixels parsed as these separate productions to be infinite.

## 3.2 INFERSSPN

Pseudocode for our algorithm, INFERSSPN, is presented in Algorithm 1. INFERSSPN is an iterative bottom-up algorithm based on graph cuts (Kolmogorov & Zabih, 2004) that provably converges to a local minimum of the energy function. In its first iteration, INFERSSPN constructs a parse tree over the full image for each production in the grammar. The parse of each terminal production is trivial to construct and simply labels each pixel as the terminal symbol. The parse for every other production $v : X \to Y_1 Y_2$ is constructed by choosing productions for $Y_1$ and $Y_2$ and fusing their corresponding parse trees to get a parse of the image as $X$. Since the grammar is non-recursive, we can construct a directed acyclic graph (DAG) containing a node for each symbol and an edge from each symbol to each constituent of each production of that symbol and then traverse this graph from the leaves (terminals) to the root (start symbol), fusing the children of each production of each symbol when we visit that symbol's node. Of course, to fuse parses of $Y_1$ and $Y_2$ into a parse of $X$, we need to choose which production of $Y_1$ (and $Y_2$) to fuse; this is done by simply choosing the production of $Y_1$ (and $Y_2$) that has the lowest energy over the current region. The best parse of the image, $\hat{t}$, now corresponds to the lowest-energy parse of all productions of the start symbol.

Further iterations of INFERSSPN improve $\hat{t}$ in a flexible manner that allows any of its productions or labelings to change, while also ensuring that its energy never increases. INFERSSPN does this by again computing parses of the full image for each production in the grammar. This time, however, when parsing a symbol $X$, INFERSSPN independently parses each region of the image that was parsed as any production of $X$ in $\hat{t}$ (none of these regions will overlap because the grammar is non-recursive) and then parses the remainder of the image *given* these parses of subregions of the image, meaning that the pixels in these other subregions are instantiated in the MRF but fixed to the labels that the subregion parses specify. The parse of the image as $X$ is then constructed as the union of these subregion parses. This procedure ensures that the energy will never increase (see Theorem 1 and Lemma 1), but also that any subtree of $\hat{t}$ can be replaced with another subtree if it results in lower energy. Figure 2b shows a simple example of updating a parse of a region as $X \to YZ$. Further, this (re)parsing of subregions can again be achieved in a single bottom-up pass through the grammar DAG, resulting in a very efficient algorithm for SSPN inference. This is because each pixel only appears in at most one subregion for any symbol, and thus only ever needs to be parsed once per production. See Algorithm 1 for more details.

## 3.3 ANALYSIS

As shown in Theorem 1, INFERSSPN always converges to a local minimum of the energy function. Similar to other graph-cut-based algorithms, such as $\alpha$-expansion (Boykov et al., 2001), IN-FERSSPN explores an exponentially large set of moves at each step, so the returned local minimum is much better than those returned by more local procedures, such as max-product belief propagation. Further, we observe convergence within a few iterations in all experiments, with the majority of the energy improvement occurring in the first iteration.

**Theorem 1.** *Given a parse (tree)* $\hat{t}$ *of* $S$ *over the entire image with energy* $E(\hat{t})$*, each iteration of* INFERSSPN *constructs a parse (tree)* $t$ *of* $S$ *over the entire image with energy* $E(t) \leq E(\hat{t})$ *and since the minimum energy of an image parse is finite,* INFERSSPN *will always converge.*

As shown in Proposition 2, each iteration of INFERSSPN takes time $O(|G|c(n))$, where $n$ is the number of pixels in the image and $c(n)$ is the complexity of the underlying graph cut algorithm used, which is low-order polynomial in the worst-case but nearly linear-time in practice (Boykov & Kolmogorov, 2004; Boykov et al., 2001).

**Proposition 2.** *Let* $c(n)$ *be the time complexity of computing a graph cut on* $n$ *pixels and* $|G|$ *be the size of the grammar defining the SSPN, then each iteration of* INFERSSPN *takes time* $O(|G|c(n))$.

---

**Algorithm 1** Compute the (approximate) MAP assignment of the SSPN variables (i.e., the productions and labelings) defined by an image and a grammar. This is equivalent to parsing the image.

---

**Input:** The image $\mathbf{x}$, a non-recursive grammar $G = (N, \Sigma, R, S, \mathbf{w})$, and (optional) input parse $\hat{t}$.
**Output:** A parse of the image, $t^*$, with energy $E(t^*, \mathbf{x}) \leq E(\hat{t}, \mathbf{x})$.

1: **function** INFERSSPN($\mathbf{x}, G, \hat{t}$)
2: $T, E \leftarrow$ empty lists of parse trees and energies, respectively, both of length $|R| + |\Sigma|$
3: **for** each terminal $Y \in \Sigma$ **do**
4: $T[Y] \leftarrow$ the trivial parse with all pixels parsed as $Y$
5: $E[Y] \leftarrow \sum_{p \in \mathbf{x}} \mathbf{w}_Y^\top \phi_p^U$
6: **while** the energy of any production of the start symbol $S$ has not converged **do**
7: **for** each symbol $X \in N$, in reverse topological order **do** *// as defined by the DAG of $G$*
8: **for** each subtree $\hat{t}_i$ of $\hat{t}$ rooted at a production $u_i$ with head $X$ **do**
9: $\mathcal{P}_i, \mathbf{y}_i \leftarrow$ the region that $\hat{t}_i$ is over and its labeling in $\hat{t}_i$ *// $\{\mathcal{P}_i\}$ are all disjoint*
10: **for** each production $v_j : X \rightarrow Y_1 Y_2$ **do** *// iterate over all productions of $X$*
11: $t_{ij}, e_{ij} \leftarrow$ FUSE($\mathcal{P}_i, \mathbf{y}_i, v_j, T$) *// parse $\mathcal{P}_i$ as $v_j$ by fusing parses of $Y_1$ and $Y_2$*
12: $\mathcal{P}_{\overline{X}} \leftarrow$ all pixels that are not in any region $\mathcal{P}_i$
13: **for** each production $v_j : X \rightarrow Y_1 Y_2$ **do** *// iterate over all productions of $X$*
14: $\mathbf{y}_{\text{rand}} \leftarrow$ a random labeling of $\mathcal{P}_{\overline{X}}$ *// use random for initialization*
15: $t_{\overline{X}}, e_{\overline{X}} \leftarrow$ FUSE($\mathcal{P}_{\overline{X}}, \mathbf{y}_{\text{rand}}, v_j, T, (\cup_i t_{ij})$) *// parse $\mathcal{P}_{\overline{X}}$ as $v_j$ given $(\cup_i t_{ij})$*
16: update lists: $T[v_j] \leftarrow (\cup_i t_{ij}) \cup t_{\overline{X}}$ and $E[v_j] \leftarrow \sum_i e_{ij} + e_{\overline{X}}$ for all $v_j$ with head $X$
17: $\hat{t}, \hat{e} \leftarrow$ the production of $S$ with the lowest energy in $E$ and its energy
18: **return** $\hat{t}, \hat{e}$

---

**Input:** A region $\mathcal{P}$, a labeling $\mathbf{y}$ of $\mathcal{P}$, a production $v : X \rightarrow Y_1 Y_2$, a list of parses $T$, and an optional parse $t_{\overline{\mathcal{P}}}$ of pixels not in $\mathcal{P}$, used to set pairwise terms of edges that are leaving $\mathcal{P}$.
**Output:** A parse tree rooted at $v$ over region $\mathcal{P}$ and the energy of that parse tree.

1: **function** FUSE($\mathcal{P}, \mathbf{y}, v, T, t_{\overline{\mathcal{P}}}$)
2: **for** each $Y_i$ with $i \in 1, 2$ **do**
3: $u_i \leftarrow$ production of $Y_i$ in $T$ with lowest energy over $\{p : y_p = Y_i\}$ given $t_{\overline{\mathcal{P}}}$
4: create submodular energy function $E(v, \mathbf{y}, \mathcal{P}, \mathbf{x})$ on $\mathcal{P}$ from $T[u_1], T[u_2]$, and $t_{\overline{\mathcal{P}}}$
5: $\mathbf{y}^v, e^v \leftarrow (\arg) \min_{\mathbf{y}} E(v, \mathbf{y}, \mathcal{P}, \mathbf{x})$ *// label each pixel in $\mathcal{P}$ as $Y_1$ or $Y_2$ using graph cuts*
6: $t^v \leftarrow$ combine $T[u_1]$ and $T[u_2]$ according to $\mathbf{y}^v$ and append $v$ as the root
7: **return** $t^v, e^v$

---

Note that a straightforward application of $\alpha$-expansion to image parsing that uses one label for every possible parse in the grammar requires an exponential number of labels in general.

INFERSSPN can be extended to productions with more than two constituents by simply replacing the internal graph cut used to fuse subtrees with a multi-label algorithm such as $\alpha$-expansion. INFERSSPN would still converge because each subtree would still never decrease in energy. An algorithm such as QPBO (Kolmogorov & Rother, 2007) could also be used, which would allow the submodularity restriction to be relaxed. Finally, running INFERSSPN on the grammar containing $k - 1$ binary productions that results from converting a grammar with a single production on $k > 2$ constituents is equivalent to running $\alpha$-expansion on the $k$ constituents.

## 4 EXPERIMENTS

We evaluated INFERSSPN by parsing images from the Stanford background dataset (SBD) using grammars with generated structure and weights inferred from the pixel labels of the images we parsed. SBD is a standard semantic segmentation dataset containing images with an average size of $320 \times 240$ pixels and a total of 8 labels. The input features we used were from the Deeplab system (Chen et al., 2015; 2016) trained on the same images used for evaluation (note that we are not evaluating learning and thus use the same features for each algorithm and evaluate on the training data in order to separate inference performance from generalization performance). We compared INFERSSPN to $\alpha$-expansion on a flat pairwise MRF and to max-product belief propagation (BP) on a multi-level (3-D) pairwise grid MRF. Details of these models are provided in the appendix. We note

that the flat encoding for $\alpha$-expansion results in a label for each path in the grammar, where there are an exponential number of such paths in the height of the grammar. However, once $\alpha$-expansion converges, its energy is within a constant factor of the global minimum energy (Boykov et al., 2001) and thus serves as a good surrogate for the true global minimum, which is intractable to compute.

We compared these algorithms by varying three different parameters: boundary strength (strength of pairwise terms), grammar height, and number of productions per nonterminal. Each grammar used for testing contained a start symbol, multiple layers of nonterminals, and a final layer of nonterminals in one-to-one correspondence with the eight terminal symbols, each of which had a single production that produces a region of pixels. The start symbol had one production for each pair of symbols in the layer below it, and the last nonterminal layer (ignoring the nonterminals for the labels) had productions for each pair of labels, distributed uniformly over this last nonterminal layer.

**Boundary strength.** Increasing the boundary strength of an MRF makes inference more challenging, as individual pixel labels cannot be easily flipped without large side effects. To test this, we constructed a grammar as above with 2 layers of nonterminals (not including the start symbol), each containing 3 nonterminal symbols with 4 binary productions to the next layer. We vary $w_v^{\text{BF}}$ for all $v$ and plot the mean average pixel accuracy returned by each algorithm (the x-axis is log-scale) in Figure 3a. INFERSSPN returns parses with almost identical accuracy (and energy) to $\alpha$-expansion. BP also returns comparable accuracies, but almost always returns invalid parses with infinite energy (if it converges at all) that contain multiple productions of the same object or a production of some symbol Y even though a pixel is labeled as symbol X.

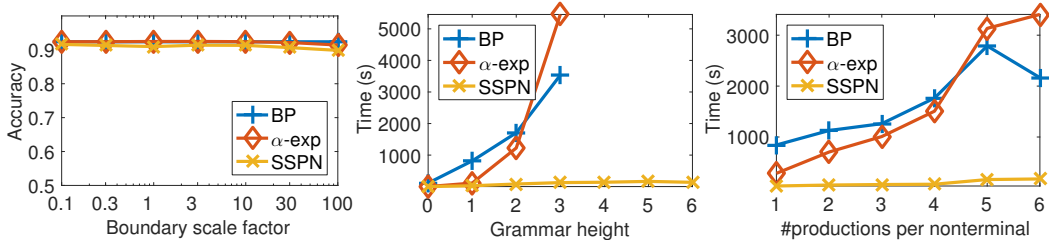

Figure 3: The mean average pixel accuracy of the returned solution and total running time for each of belief propagation, $\alpha$-expansion, and INFERSSPN when varying (a) boundary strength, (b) grammar height, and (c) number of productions. Each data point is the average value over (the same) 10 images. Missing data points indicate out of memory errors. Figures 4, 5, and 6 in the appendix show all results for each experiment.

**Grammar height.** In general, the number of paths in the grammar is exponential in its height, so the height of the grammar controls the complexity of inference and thus the difficulty of parsing images. For this experiment, we set the boundary scale factor to 10 and constructed a grammar with four nonterminals per layer, each with three binary productions to the next layer. Figure 3b shows the effect of grammar height on total inference time (to convergence or a maximum number of iterations, whichever first occurred). As expected from Proposition 2, the time taken for INFERSSPN scales linearly with the height of the grammar, which is within a constant factor of the size of the grammar when all other parameters are fixed. Similarly, inference time for both $\alpha$-expansion and BP scaled exponentially with the height of the grammar because the number of labels for both increases combinatorially. Again, the energies and corresponding accuracies achieved by INFERSSPN were nearly identical to those of $\alpha$-expansion (see Figure 5 in the appendix).

**Productions per nonterminal.** The number of paths in the grammar is also directly affected by the number of productions per symbol. For this experiment, we increased each pairwise term by a factor of 10 and constructed a grammar with 2 layers of nonterminals, each with 4 nonterminal symbols. Figure 3c shows the effect of increasing the number of productions per nonterminal, which again demonstrates that INFERSSPN is far more efficient than either $\alpha$-expansion or BP as the complexity of the grammar increases, while still finding comparable solutions (see Figure 6 in the appendix).

## 5 CONCLUSION

This paper proposed submodular sum-product networks (SSPNs), a novel extension of sum-product networks that can be understood as an instantiation of an image grammar in which all possible parses of an image over arbitrary shapes are represented. Despite this complexity, we presented

INFERSSPN, a move-making algorithm that exploits submodularity in order to find the (approximate) MAP state of an SSPN, which is equivalent to finding the (approximate) optimal parse of an image. Analytically, we showed that INFERSSPN is both very efficient – each iteration takes time linear in the size of the grammar and the complexity of one graph cut – and convergent. Empirically, we showed that INFERSSPN achieves accuracies and energies comparable to $\alpha$-expansion, which is guaranteed to return optima within a constant factor of the global optimum, while taking exponentially less time to do so.

We have begun work on learning the structure and parameters of SSPNs from data. This is a particularly promising avenue of research because many recent works have demonstrated that learning both the structure and parameters of sum-product networks from data is feasible and effective, despite the well-known difficulty of grammar induction. We also plan to apply SSPNs to additional domains, such as activity recognition, social network modeling, and probabilistic knowledge bases.

## ACKNOWLEDGMENTS

AF would like to thank Robert Gens and Rahul Kidambi for useful discussions and insights, and Gena Barnabee for assisting with Figure 1 and for feedback on this document. This research was partly funded by ONR grant N00014-16-1-2697 and AFRL contract FA8750-13-2-0019. The views and conclusions contained in this document are those of the authors and should not be interpreted as necessarily representing the official policies, either expressed or implied, of ONR, AFRL, or the United States Government.

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

## A    PROOFS

**Proposition 1.** *The energy $E(v, t_1, t_2, \mathbf{y}^v)$ of the fusion of parse trees $t_1, t_2$ over region $\mathcal{P}$ with head symbols $Y_1, Y_2$ for a production $v : X \to Y_1 Y_2$ is submodular.*

*Proof.* $E(v, t_1, t_2)$ is submodular as long as $2 \cdot \theta_{pq}^v(Y_1, Y_2) \geq \theta_{pq}^{t_1} + \theta_{pq}^{t_2}$, which is true by construction, since $\theta_{pq}^v(y_p^v, y_q^v) \geq \theta_{pq}^c(y_p^c, y_q^c)$ for $c$ any possible descendant of $v$ and for all labelings. $\quad\square$

**Lemma 2.** *Given a labeling $\mathbf{y}^v$ which fuses parse trees $t_1, t_2$ into $t$ with root production $v$, energy $E(t, \mathcal{P}) = E(v, t_1, t_2, \mathbf{y}^v)$, and subtree regions $\mathcal{P}_1 \cap \mathcal{P}_2 = \emptyset$ defined by $\mathbf{y}^v$, then any improvement $\Delta$ in $E(t_1, \mathcal{P}_1)$ also improves $E(t, \mathcal{P})$ by at least $\Delta$, regardless of any change in $E(t_1, \mathcal{P}\backslash\mathcal{P}_1)$.*

*Proof.* Since the optimal fusion can be found exactly, and the energy of the current labeling $\mathbf{y}^v$ has improved by $\Delta$, the optimal fusion will have improved by at least $\Delta$. $\quad\square$

**Proposition 2.** *Let $c(n)$ be the time complexity of computing a graph cut on $n$ pixels and $|G|$ be the size of the grammar defining the SSPN, then each iteration of INFERSSPN takes time $O(|G|c(n))$.*

*Proof.* Let $k$ be the number of productions per nonterminal symbol and $N$ be the nonterminals. For each nonterminal, FUSE is called $k$ times for each region and once for the remainder of the pixels. FUSE itself has complexity $O(|\mathcal{P}| + c(|\mathcal{P}|)) = O(c(|\mathcal{P}|))$ when called with region $\mathcal{P}$. However, in INFERSSPN each pixel is processed only once for each symbol because no regions overlap, so the worst-case complexity occurs when each symbol has only one region, and thus the total complexity of each iteration of INFERSSPN is $O(|N|k \cdot c(n)) = O(|G|c(n))$. $\quad\square$

**Theorem 2.** *Given a parse (tree) $\hat{t}$ of $S$ over the entire image with energy $E(\hat{t})$, each iteration of INFERSSPN constructs a parse (tree) $t$ of $S$ over the entire image with energy $E(t) \leq E(\hat{t})$, and since the minimum energy of an image parse is finite, INFERSSPN will always converge.*

*Proof.* We will prove by induction that for all nodes $n_i \in \hat{t}$ with corresponding subtree $\hat{t}_i$, region $\mathcal{P}_i$, production $v_i : X \to Y_1 Y_2$ and child subtrees $\hat{t}_1, \hat{t}_2$, that $E(t_i) \leq E(\hat{t}_i)$ after one iteration for all $t_i = T[v_i] \cap \mathcal{P}_i$. Since this holds for every production of $S$ over the image, this proves the claim.
**Base case.** When $\hat{t}_i$ is the subtree with region $\mathcal{P}_i$ and production $v_i : X \to Y$ containing only a single terminal child, then by definition $t_i = T[v_i] \cap \mathcal{P}_i = \hat{t}_i$ because terminal parses do not change given the same region. Thus, $E(t_i) = E(\hat{t}_i)$ and the claim holds.
**Induction step.** Let $v_i : X \to Y_1 Y_2$ be the production for a node in $\hat{t}_i$ with subtrees $\hat{t}_1, \hat{t}_2$ over regions $\mathcal{P}_1, \mathcal{P}_2$, respectively, such that $\mathcal{P}_1 \cup \mathcal{P}_2 = \mathcal{P}_i$ and $\mathcal{P}_1 \cap \mathcal{P}_2 = \emptyset$, and suppose that for all productions $u_{1j}$ with head $Y_1$ and all productions $u_{2k}$ with head $Y_2$ and corresponding parse trees $t_{1j} = T[u_{1j}] \cap \mathcal{P}_1$ and $t_{2k} = T[u_{2k}] \cap \mathcal{P}_2$, respectively, that $E(t_{1j}) \leq E(\hat{t}_{1j})$ and $E(t_{2k}) \leq E(\hat{t}_{2k})$. Now, when FUSE is called on region $\mathcal{P}_1$ it will choose the subtrees $t_{1j}^* : j = \arg\min_j E(t_{1j}, \mathcal{P}_1)$, and $t_{2k}^* : k = \arg\min_k E(t_{2k}, \mathcal{P}_2)$ and fuse these into $t_i'$ over $\mathcal{P}$. However, from Lemma 1, we know that $t_i$ could at the very least simply reuse the labeling $\mathbf{y}^v$ that partitions $\mathcal{P}$ into $\mathcal{P}_1, \mathcal{P}_2$ and in doing so return a tree $t_i'$ with energy $E(t_i') \leq E(\hat{t}_i)$, because each of its subtrees over their same regions has lower (or equal) energy to those in $\hat{t}$. Finally, since $t_i'$ is computed independently of any other trees for region $\mathcal{P}$ and then placed into $T[v_i]$ as a union of other trees, then $t_i = T[v_i] \cap \mathcal{P} = t_i'$, and the claim follows. $\quad\square$

## B    ADDITIONAL EXPERIMENTAL RESULTS AND DETAILS

We compared INFERSSPN to running $\alpha$-expansion on a flat pairwise MRF and to max-product belief propagation over a multi-level (3-D) pairwise grid MRF. Each label of the flat MRF corresponds to a possible path in the grammar from the start symbol to a production to one of its constituent symbols, etc, until reaching a terminal. In general, the number of such paths is exponential in the height of the grammar. The unary terms are the sum of unary terms along the path and the pairwise term for a pair of labels is the pairwise term of the first production at which their constituents differ. For any two labels with paths that choose a different production of the same symbol (and have the same path from the start symbol) we assign infinite cost to enforce the restriction that an object can only have a single production of it into constituents. Note that after convergence $\alpha$-expansion is

guaranteed to be within a constant factor of the global minimum energy (Boykov et al., 2001) and thus serves as a good surrogate for the true global minimum, which is intractable to compute. The multi-layer MRF is constructed similarly. The number of levels in the MRF is equal to the height of the DAG corresponding to the grammar used. The labels at a particular level of the MRF are all (production, constituent) pairs that can occur at this height in the grammar. The pairwise term between the same pixel in two levels is 0 when the parent label's constituent equals the child label's production head, and $\infty$ otherwise. Pairwise terms within a layer are defined as in the flat MRF with infinite cost for incompatible labels (i.e., two neighboring productions of the same symbol), unless two copies of that nonterminal could be produced at that level by the grammar.

All experiments were run on the same computer running an Intel Core i7-5960X with 8 cores and 128MB of RAM. Each algorithm was limited to a single thread.

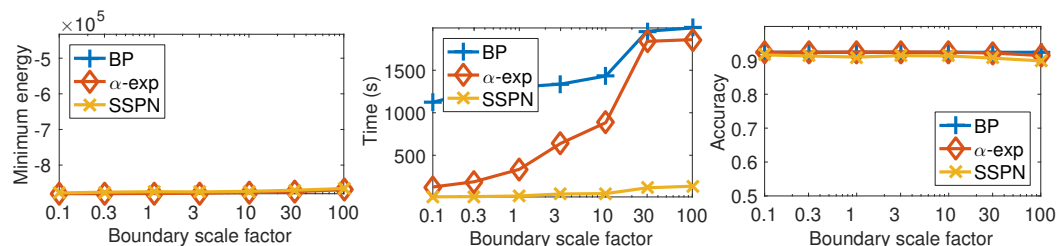

Figure 4: The (a) best energy, (b) total running time, and (c) resulting semantic segmentation accuracy (mean average pixel accuracy) for belief propagation, $\alpha$-expansion, and INFERSSPN when varying boundary strength. Each data point is the average value over (the same) 10 images. Missing data points indicate that an algorithm ran out of memory (middle and right) or returned infinite energy (left).

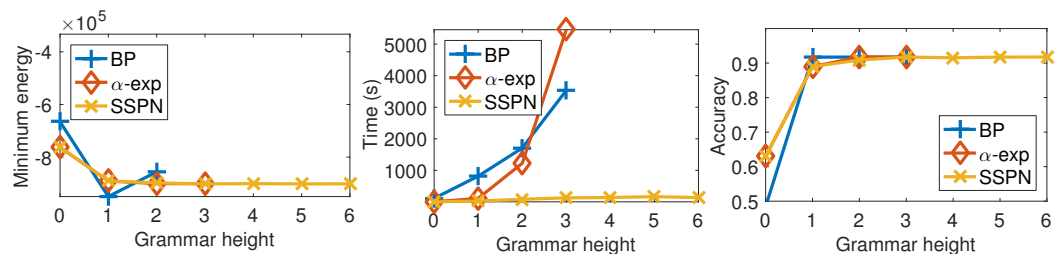

Figure 5: The (a) best energy, (b) total running time, and (c) resulting semantic segmentation accuracy (mean average pixel accuracy) for belief propagation, $\alpha$-expansion, and INFERSSPN when varying grammar height. Each data point is the average value over (the same) 10 images. Missing data points indicate that an algorithm ran out of memory (middle and right) or returned infinite energy (left). Low accuracies for grammar height 0 are a result of the grammar being insufficiently expressive.

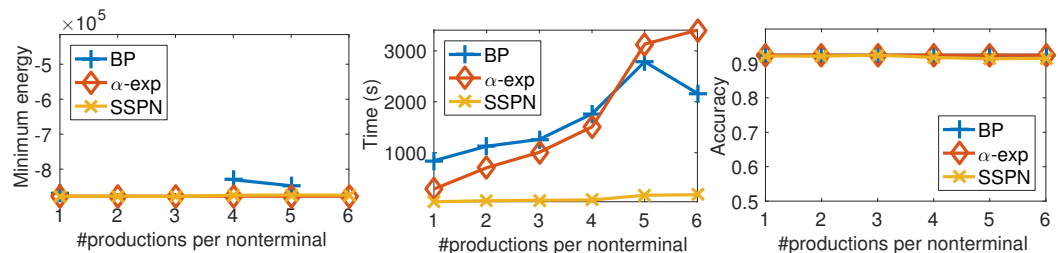

Figure 6: The (a) best energy, (b) total running time, and (c) resulting semantic segmentation accuracy (mean average pixel accuracy) for belief propagation, $\alpha$-expansion, and INFERSSPN when varying grammar height. Each data point is the average value over (the same) 10 images. Missing data points indicate that an algorithm ran out of memory (middle and right) or returned infinite energy (left).

