# Peer review of "Submodular Sum-product Networks for Scene Understanding"

_ICLR 2017 — rejected_

[Official Review · AnonReviewer3 · rating 4 · confidence 3 · 16 Dec 2016]
**Interesting idea that needs to be fully developed and evaluated**

This paper is about submodular sum product networks applied to scene understanding. SPNs have shown great success in deep linear models since the work of Poon 2011.  The authors propose an extension to the initial SPNs model to be submodular, introducing submodular unary and pairwise potentials.  The authors propose a new inference algorithm. The authors evaluated their results on Stanford Background Dataset and compared against multiple baselines.

Pros:
+ New formulation of SPNs 
+ New inference algorithm

Cons:
- The authors did not discuss how the SSPN structure is learned and how the generative process chooses the a symbol (operation) at each level)
- The evaluations is lacking. The authors only showed results on their own approach and baselines, leaving out every other approach. Evaluations could have been also done on BSD for regular image segmentation (hierarchical segmentation). 

The idea is great, however, the paper needs more work to be published.  I would also recommend for the authors to include more details about their approach and present a full paper with extended experiments and full learning approach.

[Official Review · AnonReviewer2 · rating 4 · confidence 3 · 16 Dec 2016]
**Good paper, but not a good fit for ICLR**

This paper develops Submodular Sum Product Networks (SSPNs) and
an efficient inference algorithm for approximately computing the
most probable labeling of variables in the model. The main
application in the paper is on scene parsing. In this context,
SSPNs define an energy function with a grammar component for
representing a hierarchy of labels and an MRF for encoding
smoothness of labels over space. To perform inference, the
authors develop a move-making algorithm, somewhat in the spirit
of fusion moves (Lempitsky et al., 2010) that repeatedly improves
a solution by considering a large neighborhood of alternative segmentations
and solving an optimization problem to choose the best neighbor.
Empirical results show that the proposed algorithm achieves better
energy that belief propagation of alpha expansion and is much faster.

This is generally a well-executed paper. The model is interesting
and clearly defined, the algorithm is well presented with proper
analysis of the relevant runtimes and guarantees on the
behavior. Overall, the algorithm seems effective at minimizing
the energy of SSPN models.

Having said that, I don't think this paper is a great fit for
ICLR. The model is even somewhat to the antithesis of the idea of
learning representations, in that a highly structured form of
energy function is asserted by the human modeller, and then
inference is performed. I don't see the connection to learning
representations. One additional issue is that while the proposed
algorithm is faster than alternatives, the times are still on the
order of 1-287 seconds per image, which means that the
applicability of this method (as is) to something like training
ConvNets is limited.

Finally, there is no attempt to argue that the model produces
better segmentations than alternative models. The only
evaluations in the paper are on energy values achieved and on
training data.

So overall I think this is a good paper that should be published
at a good machine learning conference, but I don't think ICLR is
the right fit.

[Official Review · AnonReviewer1 · rating 5 · confidence 4 · 17 Dec 2016]
**My thoughts**

The paper discusses sub modular sum-product networks as a tractable extension for classical sum-product networks. The proposed approach is evaluated on semantic segmentation tasks and some early promising results are provided.

Summary:
———
I think the paper presents a compelling technique for hierarchical reasoning in MRFs but the experimental results are not yet convincing. Moreover the writing is confusing at times. See below for details.

Quality: I think some of the techniques could be described more carefully to better convey the intuition.
Clarity: Some of the derivations and intuitions could be explained in more detail.
Originality: The suggested idea is great.
Significance: Since the experimental setup is somewhat limited according to my opinion, significance is hard to judge at this point in time.

Detailed comments:
———
1. I think the clarity of the paper would benefit significantly from fixes to inaccuracies. E.g., \alpha-expansion and belief propagation are not `scene-understanding algorithms’ but rather approaches for optimizing energy functions. Computing the MAP state of an SSPN in time sub-linear in the network size seems counterintuitive because it means we are not allowed to visit all the nodes in the network. The term `deep probabilistic model’ should probably be defined. The paper states that InferSSPN computes `the approximate MAP state of the SSPN (equivalently, the optimal parse of the image)’ and I’m wondering how the `approximate MAP state' can be optimal. Etc.

2. Albeit being formulated for scene understanding tasks, no experiments demonstrate the obtained results of the proposed technique. To assess the applicability of the proposed approach a more detailed analysis is required. More specifically, the technique is evaluated on a subset of images which makes comparison to any other approach impossible. According to my opinion, either a conclusive experimental evaluation using, e.g., IoU metric should be given in the paper, or a comparison to publicly available results is possible.

3. To simplify the understanding of the paper a more intuitive high-level description is desirable. Maybe the authors can even provide an intuitive visualization of their approach.

[Author Response · Abram L. Friesen · 14 Jan 2017]
**General response to concern about our empirical results**

The main concern raised in each review is the lack of a direct comparison of SSPNs to existing semantic segmentation benchmarks. We address this point here and then respond to each reviewer's other points directly. We’ve also made significant revisions to the structure and writing of the paper, particularly in the inference section, which we hope more clearly explains the intuitions behind InferSSPN.

Despite not having comparisons on semantic segmentation benchmarks, SSPNs are a significant step forward in the state-of-the-art on scene understanding. SSPNs are (provably) more expressive than (and contain as a special case) MRFs and SPNs as commonly used for scene understanding and semantic segmentation, meaning that SSPNs are a richer model class than those already in use. Despite this increase in expressivity and complexity, InferSSPN is a very efficient (approximate) inference algorithm for the problem of parsing an image with respect to a grammar, which has been addressed by many previous works, but their approaches have all been very restrictive or inefficient. InferSSPN provably achieves low-order-polynomial time complexity for a problem with a state space that has size exponential in both the number of pixels and the height of the grammar. While the first version of the paper did not include accuracy results, as we thought the energy results were sufficient, we’ve since updated the paper to include all accuracy, energy, and time complexity results for all test cases. These show empirically that InferSSPN provides exponential improvements in time-complexity with no loss in accuracy relative to alpha-expansion, which provably returns local optima that are within a constant factor of the global optimum.

Further, while we have characterized SSPNs in terms of scene understanding, they are as broadly applicable as SPNs and MRFs, and scene understanding is just the initial application we chose to focus on. Future applications may include activity recognition or social network modeling. Nonetheless, this paper presents a powerful new representation for efficiently modeling and reasoning about scenes at multiple levels of abstraction and is well within the scope of ICLR, which includes “hierarchical models” and “applications to vision” in its list of relevant topics.

Learning SSPNs is important – both for evaluating SSPNs on existing semantic segmentation benchmarks and for their adoption in general – and is something that we are already working on. Unfortunately, there do not yet exist any image grammars or relevant datasets with which to train SSPNs, so learning requires either creating such datasets or learning with a large number of hidden variables. This is ongoing and future work. However, we believe that one benefit of our contributions will be to spur the creation of such datasets and open up new research problems where reasoning about multiple levels of abstraction in scenes and higher-level relationships between objects is crucial. Finally, it’s common in AI and ML for new representations to be introduced well before learning algorithms are properly developed for them. Notable cases include graphical models and previous work on image grammars, and this paper does the same for SSPNs.

[Final Decision · Program Chairs · 06 Feb 2017]
**ICLR committee final decision**

This paper was reviewed by three experts. While they all find merits in the paper (interesting new model class SSPN, new MAP inference algorithm), they all consistently point to deficiencies in the current manuscript (lack of parameter learning, emphasis on evaluation on energies, lack of improvements in accuracy). 
 
 One problem that (I believe) is that manuscript as it stands makes neither a compelling impact on the chosen application (semantic segmentation) nor does it convincing establish the broad applicability of the proposed model (how do I run SSPNs on activity recognition or social network modeling). 
 
 To be clear, we all agree that there is promising content here. However, I agree with the reviewers that the significance has not been established.